# Simultaneous multicopter-based air sampling and sensing of meteorological variables

Caroline Brosy[1], Karina Krampf[l], Matthias Zeeman[1], Benjamin Wolf[l], Wolfgang Junkermann[1], Klaus Schäfer[1], Stefan Emeis[1], Harald Kunstmann[1,2]

[1]Institute of Meteorology and Climate Research (IMK-IFU), Karlsruhe Institute of Technology, Garmisch-Partenkirchen, 82467, Germany
[2]Institute of Geography, University of Augsburg, Augsburg, 86159, Germany

*Correspondence to*: Caroline Brosy (caroline.brosy@kit.edu)

**Abstract.** The state and composition of the lowest part of the planetary boundary layer (PBL), i.e., the atmospheric surface
layer (SL), reflects the interactions of external forcing, land surface, vegetation, human influence and the atmosphere. Vertical profiles of atmospheric variables in the SL at high spatial (meters) and temporal (1 Hz and better) resolution increase our understanding of these interactions, but are still challenging to measure appropriately. Traditional ground-based observations include towers that often cover only few measurement heights on a fixed location. At the same time, most remote sensing techniques and aircraft measurements have limitations to achieve sufficient detail close to the ground (up to
50 m). Vertical and horizontal transects of the PBL can be complemented by unmanned aerial vehicles (UAV). Our aim in this case study is to assess the use of a multicopter-type UAV for the spatial sampling of air and simultaneously the sensing of meteorological variables for the study of the surface exchange processes. To this end, a UAV was equipped with onboard air temperature and humidity sensors, while wind conditions were determined from the UAV's flight control sensors. Further, the UAV was used to systematically change the location of a sample inlet connected to a sample tube, allowing the
observation of methane abundance using a ground-based analyzer. Vertical methane gradients of about 0.3 ppm were found during stable atmospheric conditions. Our results showed that both methane and meteorological conditions were in agreement with other observations at the site during the ScaleX-2015 campaign. The multicopter-type UAV was capable of simultaneous in situ sensing of meteorological state variables and sampling of air up to 50 m above the surface, which extended the vertical profile height of existing tower-based infrastructure by a factor of five.

**1 Introduction**

The planetary boundary layer (PBL) is the lowest part of the atmosphere directly influenced by the Earth's surface and reflects interactions between land surface, vegetation, human activities and the atmosphere (Stull, 1988). Since mixing processes and transport or the lack of those affect trace gas and aerosol distributions in the atmosphere on all scales, vertical profiles provide more detailed information which have to be accounted for when dealing with emission and flux estimations
(Worden et al., 2012).

Well-known in situ platforms for the measurement of vertical profiles of atmospheric variables in the PBL are towers, (tethered) balloons and radiosondes (Konrad et al., 1970). The operation of towers is fixed to a certain location and the vertical information is limited to the height of the tower as well as to discrete levels at the tower. However, towers provide continuous recording of the investigated variables and are routinely used (e.g. Sasakawa et al., 2010; Wang et al., 2013; Andrews et al., 2014). With radiosondes, balloons or kites, information of meteorological conditions can be acquired for an extended vertical range, but these systems are expensive and the location of the vertical profiles is dependent on atmospheric conditions. Nevertheless, mobile and temporary applications are possible. Research aircraft and satellites cover large areas in the range of kilometers within a short time span, but their operation close to the ground is still challenging (Velasco et al., 2008; Martin et al., 2011). Considering ground-based remote sensing methods, data of vertical profiles from low altitudes up to about 50 m above ground level (a.g.l.) are hardly usable (e.g. acoustic instruments), but possible with LIDARs applying certain scan patterns with low elevation angles at the position of such an instrument (Emeis, et al., 2009; Banta et al., 2013; Korhonen et al., 2014; Hammann et al., 2015).

From the 1970s on, UAVs were used for atmospheric research, for example for convective processes (Konrad et al., 1970; Rennó and Williams, 1995) and weather forecast (Holland et al., 1992; McGeer and Holland, 1993), as well as for vertical sounding of the planetary boundary layer (Egger et al., 2002; Soddell et al., 2004; Spiess et al., 2007). In recent years, unmanned aerial vehicles (UAVs) became increasingly used as flying platforms for measurements in atmospheric research for both vertical and horizontal applications (Villa et al., 2016). Martin et al. (2011) demonstrated the utilization of fixed-wing UAVs for measurements of meteorological variables, i.e. air temperature, humidity and wind, up to 1600 m above ground level (a.g.l.). In addition, de Boer et al. (2016) implemented radiation and aerosol size distributions sensors, Altstädter et al. (2015) focused on ultrafine particles and Båserud et al. (2016) showed the possibility of turbulence measurements. Nathan et al. (2015) measured methane with an in situ sensor flying around a compressor station to calculate its emissions. The importance of knowing both meteorological conditions and methane (or aerosols, particulate matter, etc.) was highlighted in previous studies (e.g. Bamberger et al., 2014; Mathieu et al., 2015). Fixed-wing systems can cover a vertical and horizontal range of several kilometers and therefore, they are suitable for investigations throughout the boundary layer. Multicopters offer flexible maneuverability at low flight speed and the possibility of hovering (i.e. no horizontal movement). Their applications include meteorological and air quality measurements, e.g. particulate matter (Alvarado et al., 2015) or air samples for analyses of chemical composition (Chang et al. 2016), but on a smaller scale of several hundreds of meters. In addition, Neumann and Bartholomai (2015) and Palomaki et al. (2017) showed that the onboard flight control sensors can be used to derive wind estimates from a multicopter's attitude control data. Although small and lightweight methane sensors are available (Berman et al., 2012; Khan et al., 2012), current model multicopters with a takeoff weight below 5 kg still require further miniaturization of the sensors. As a consequence, mobile investigations of vertically resolved profiles of greenhouse gases in combination with information about atmospheric state variables are not conventionally applied yet.

Methane ($CH_4$) is the second most important greenhouse gas with regard to global warming and has a global warming potential 20 times that of carbon dioxide (Forster et al., 2007). Its current concentration is more than twice as much as before preindustrial times (Kirschke et al., 2013; Saunois et al., 2016). While the global budget is well known, this is not the case for regional to local scales, especially vertical distributions (Dlugokencky et al., 2011). Using tethered balloons, Choularton et al. (1995), Beswick et al. (1998) and Stieger et al. (2015) investigated the vertical methane distribution within the nocturnal boundary layer (NBL) by pulling up a sampling tube.

In this case study we aim to assess the feasibility of a multicopter-type UAV approach to detach methane and meteorological measurements at a tower by pulling up a tube with a multicopter weighing below 5 kg. Both methane and atmospheric state are important to analyze the vertical methane distribution focusing on stable atmospheric conditions above a typical agricultural setting in the foothills of the Bavarian Alps. The nighttime is of particular interest because turbulent mixing is low and vertical methane gradients can develop.

## 2 Methodology

### 2.1 Site description and instrumentation

The first experiments with our multicopter took place at the measurement site Fendt (DE-Fen) of the TERENO-preAlpine (TERrestrial ENvironmental Observatory) observatory (Zacharias et al., 2011) during the ScaleX campaign (Wolf et al., 2016, in press) in June and July 2015. This intensive campaign aimed to address atmosphere-land surface interactions across different scales with both measurements and modeling. As a long-term TERENO site, DE-Fen is equipped with automated instrumentation and continuous data availability already for several years. During ScaleX, measurements of energy, water and greenhouse gas fluxes were extended in time and spatial resolution to investigate spatial patterns and vertical gradients to obtain three-dimensional and more detailed information.

DE-Fen (47.832 °N, 11.062 °E, 600 m above sea level (a.s.l.)) is located in a north-south oriented valley in the foothills of the Bavarian Alps in southern Germany (Fig. 1). While the surface is relatively flat towards the east about 300 m to the west a steep forested slope of about 100 to 130 m borders the grassland in the valley. Thus, orographical winds and diurnal wind systems favor northerly and southerly directions with occasional easterly or northeasterly components. Westerly winds are normally associated with orographic turbulence. Prevailing land use is grassland with sporadic croplands. Further details on climate characteristics of the region can be found in Kunstmann et al. (2004, 2006).

The site is equipped, among other instruments, with a permanent eddy-covariance (EC) station for carbon dioxide, water vapor and energy flux measurements (Mauder et al., 2013; Zeeman et al., 2017). An overview about the location of instruments is given in Fig. 1. During the campaign, a radio acoustic sounding system (SODAR-RASS, Metek GmbH, Elmshorn, Germany) was installed on the east side of the area. The SODAR-RASS consists of a SODAR for the wind measurement with an acoustic signal and two RADAR antennas for measurements of vertical profiles of air temperature (Emeis et al., 2009). The temporal resolution is 10 min with a range between 40 m to 650 m a.g.l. and a vertical resolution of

20 m. In addition, vertical profiles of wind direction and speed were determined at the intercept of three simultaneously scanning Doppler wind-LIDAR systems (model Stream Line, Halo Photonics Ltd, Worcester, UK) as a so-called 'virtual tower', in 1 min and 18 m intervals and up to approximately 800 m a.s.l.. Methane mixing ratios were determined using a cavity ring down (CRD) spectrometer (G2508, Picarro Inc., Santa Clara, CA, USA) with an accuracy of < 0.007 ppm. The instrument was installed close to a 10 m tower equipped with wind speed and direction measurements (CSAT3, Campbell Scientific Ltd., Bremen, Germany and WindMaster 3D, Gill Instruments, Lymington, Hampshire, UK) and sample air inlets at 1, 5 and 10 m height. The three sampling lines (stainless steel, 3.2 mm outer diameter, 1.2 mm inner diameter) were flushed continuously with ambient air and a custom built system of solenoid valves connected one sampling line to the CRD spectrometer every 75 s. Those measurements were complemented with UAV-based measurements being explained in Sect. 2.2 and 2.4.

## 2.2 Multicopter and its instrumentation

The multicopter used in this study was a commercially available hexacopter DJI F550 Flame Wheel (DJI Innovations, Shenzhen, China) with dimensions of 55 cm x 55 cm x 30 cm and a frame weight of 1.3 kg including motors, propellers, autopilot and electronics (Fig. 2). It was equipped with a Pixhawk (3DR, Berkley, USA) autopilot for stabilized and autonomous flights. The autopilot contains a 3D accelerometer, gyroscope, magnetometer and barometer for position control as well as an external GPS (LEA-6 u-blox 6, u-blox, Thalwil, Switzerland) for autonomous flying. All data were logged onboard, attitude angles as well as motor output at 10 Hz, the accelerometer and gyroscope data at 50 Hz and GPS at 5 Hz. Additionally, a remote receiver was installed onboard for manual flying with remote control. The takeoff weight of 2 kg led to a flying time of approximately 10 min with a ground speed of 5 m s$^{-1}$. In case of a communication loss of the remote control, GPS signal or low battery status a pre-programmed fail-safe mode took over the control and initiated the landing. The open-source software Mission Planner was used for ground control to transmit and display important flight data (e.g. height, horizontal and vertical speed, battery capacity, position) during the flights. For night flights, bright LEDs were mounted on the landing gear of the multicopter for visibility and the identification of its orientation.

This kind of UAV with a weight below 5 kg was chosen to fly with general flight permission from the Bavarian aviation authority, independent on area, altitude above ground and time of the day in the uncontrolled air space.

For vertical methane investigations close to the tower, a 40 cm long aluminum tube (3.2 mm outer diameter, 1.2 mm inner diameter) was installed on the multicopter with the inlet about 30 cm above the propellers. This was attached airtight to an additional sampling line (PTFE, 3.2 mm outer diameter, 2 mm inner diameter, 70 m long) and was connecting the CRD spectrometer and the multicopter. The 70 m sample line was flushed at a flow rate of 350 sccm min$^{-1}$ (calibrated for 0 °C and 1013.25 hPa) of which 200 sccm min$^{-1}$ were drawn by the CRD analyzer. This resulted in a residence time of approximately 38 s in the tube. At 50 m length, the tube was an additional payload of 650 g. Thus, the maximum ascent height was limited by the payload capacity of the multicopter.

A fast thermocouple was installed for high time resolution air temperature measurements. The used thermocouple was a butt welded type K (CHROMEGA®/ALOMEGA® CHAL-003, OMEGA, Stamford, CT, USA), one wire chromium nickel alloy and the other constantan, both with a diameter of 0.08 mm. Its measurement range was 0 ° to 60 °C with an output voltage of 50 mV per °C. The response time was better than 1 Hz in calm air with an accuracy of ±0.1 °C. Calibration against a reference thermometer was done in the lab. Data were logged at 10 Hz. These data were also used together with pressure data from the autopilot for potential temperature ($T_{pot}$) calculations to get information about the stability of the atmosphere. The used pressure sensor is a MS5611-01BA03 (AMSYS, Mainz, Germany) and is able to resolve an altitude of 10 cm corresponding to a precision of about ±0.02 hPa.

## 2.3 Wind estimation

Multicopters move through the air by setting a tilt angle ($\gamma$) towards the flying direction with the magnitude of tilt angle roughly proportional to speed. This angle is also changing for compensation of wind variations during the flight. Therefore, without using an additional sensor for wind measurements, estimation of both horizontal wind speed and direction was possible with onboard sensors for the vehicle's attitude control by measuring the pitch (for- and backwards), roll (left and right) and yaw (orientation to north) angles. In contrast to an aircraft, which is controlled by setting a true air speed, a multicopter is flying with a given ground speed resulting in a varying true air speed.

This relationship is shown in the wind triangle (Fig. 3). The ground (G) vector represents the speed and direction of the multicopter's movement determined by the GPS, while the true air speed (TAS) vector represents for the actual speed and direction the multicopter is heading to. The deviation of G and TAS is caused by the wind. Assuming hovering, the tilt angle is only a result of the wind and the TAS vector is contrary to the ground vector. Consequently, in the easiest case the direction of TAS represents the horizontal wind direction and the length of the TAS vector the horizontal wind speed.

Equations applied for the wind calculation are based on Neumann and Bartholomai (2015) and explained in detail there. In this study, only vertical flights were investigated. First, the multicopter's tilt angle γ was calculated from roll and pitch angles and then projected to the xy-plane, which results in the true air speed vector. Then, its direction was calculated relative to the viewing direction of the multicopter (yaw angle (ψ)) and is given by the angle λ. TAS direction and simultaneous wind direction was determined by the sum of ψ and λ in case the TAS vector is on the right side of the viewing direction ($[\psi, \psi + 180°]$). In the other case, this sum was subtracted from 360° and in both cases the result has to be within 0° and 360°. Finally, the calculated tilt angle was inserted into a regression function to get the corresponding true air speed. The length of the TAS vector represents the wind speed.

Neumann and Bartholomai (2015) used wind tunnel experiments to determine the regression function. In contrast, in our approach the length of the TAS vector was determined by relating tilt angles to specific true air speeds during different flight experiments. The assumption was that without wind the true air speed corresponds to the flight speed measured with the GPS (GPS speed or ground speed), which has an accuracy of 0.1 m s$^{-1}$. The multicopter's tilt angle was calculated by using pitch and roll angles. Their accuracy was better than 0.1°. Using racetrack flights, the regression function was experimentally

determined during calm wind conditions with wind speeds below 1 m s$^{-1}$. The track had a length of 120 m and had been flown six times on average for several ground speeds between 2 m s$^{-1}$ and 8 m s$^{-1}$. While the ground speed was kept constant by the GPS ($< \pm 0.2$ m s$^{-1}$), the variability of the assigned tilt angle was dependent on atmospheric conditions. To avoid an offset in the regression function the multicopter was balanced out. The resulting regression function is shown in Fig. 4 with the following Eq. (1):

$$TAS = 0.9743 * \gamma^{0.8817} \tag{1}$$

The root-mean-square-error (RMSE) of TAS determination was $\pm 0.3$ m s$^{-1}$. Based on this error for TAS, the RMSE of the tilt angle was $\pm 0.4°$, which is similar to the one of Neumann and Bartholomai (2015). This mean error of TAS leads to a higher relative error for low wind speeds than for higher wind speeds.

With this equation, horizontal wind speed and wind direction were estimated from 1 Hz data and were averaged with a moving window over 10 s for further smoothing. To determine the inaccuracy caused by a wind speed up to 1 m s$^{-1}$ during the experimental flights, the variability of the tilt angle was analyzed during hovering under calm wind conditions (<1 m s$^{-1}$). This led to an uncertainty of $0.7° \pm 0.3°$ corresponding to a true air speed of 0.7 m s$^{-1}$ $\pm 0.3$ m s$^{-1}$, which resulted in an overall accuracy of TAS estimation of 0.7 m s$^{-1}$ $\pm 0.6$ m s$^{-1}$.

## 2.4 Flight strategies

To demonstrate the functionality of the wind estimation based on the attitude control sensors of the multicopter, a comparison was done to a 3D ultrasonic anemometer (uSonic3, Metek GmbH, Elmshorn, Germany) installed at a 9 m tower having an accuracy of 0.1 m s$^{-1}$ and 2° at 5 m s$^{-1}$, respectively. During windy conditions (3–5 m s$^{-1}$) the multicopter was hovering for 5 min close to the tower at a distance of approximately 5 m. This horizontal distance as well as the 9 m height of the measurements ensured that the multicopter's downwash neither had an influence on the multicopter itself nor on the anemometer. For calm wind conditions, influences of the downwash were detected up to 5–6 m a.g.l.

In addition, vertical wind profiles were compared to other instruments such as LIDAR and SODAR. The EC station was used as continuous time series information close to the ground. Reaching a height of 150 m a.g.l. with the multicopter, the range comparable to other instruments was about 100 m. For these flights, the vertical speed was set to 1.5 m s$^{-1}$.

For methane measurements, the additional sampling line was attached to the multicopter and the spectrometer and was raised up to heights of 10, 25 and 50 m a.g.l. A hover time of 60 s at each level was included to get an averaged value. The pattern was repeated every 15 min. This led to a rotation of 5 min measurements with the multicopter and 10 min measurements at the tower at 1 m and 10 m a.g.l. For analysis, only the ascent data were used from the flights because there was no hovering during the descent. In addition, this strategy ensured that the multicopter did not mix the air before flying through. Alvarado et al. (2017) experimentally determined a distance of 40–45 cm above the multicopter, where the influence of the rotors to air speed decreases significantly. So, the methane mixing ratio is actually not a point measurement but valid for a volume.

While most of the flights were done above the grassland site south-west of the EC station as shown in Fig. 1, the flights including methane measurements took place close to the methane tower in the south-east of the investigation area.

Time is given in UTC which corresponds to CEST-2.

## 3 Results

### 3.1 Wind estimation

Information about the accuracy of the wind estimation was determined while hovering next to an ultrasonic anemometer
with a distance of about 5 m (Fig. 5). The multicopter derived wind direction showed a standard deviation of ±11.1° and
±0.7 m s$^{-1}$ for wind speed within a hovering time of 5 min. During the same time, the anemometer's wind direction varied by
±10.6° and wind speed by ±1 m s$^{-1}$. The difference between the multicopter and tower measurements both averaged over
5 min was 7.7° and 0.3 m s$^{-1}$. For both time series the 10 s moving average was applied resulting in a RMSE between
multicopter and tower of 14.5° and 0.7 m s$^{-1}$, respectively. Both changes in wind speed and direction could be captured by
the multicopter. The highest deviation was between 150 s and 200 s with differences of about 30° and 2 m s$^{-1}$, respectively
(see Fig. 5). Since the volume of the multicopter is larger compared to the measurement path of the sonic anemometer, the
multicopter does not react to the small turbulent elements, the so-called eddies, and therefore cannot capture the full range of
wind speed. In addition, the multicopter has inertia due to its weight. Consequently, the wind speed deviations measured by
the multicopter should not be used as information about atmospheric turbulence.

In addition to the side-by-side measurements, wind estimation from vertical profiles was compared to LIDAR and SODAR
measurements as well as EC station data for near ground information (Fig. 6). Both LIDAR and EC station data (both 1 min
time resolution) are shown for the time around the vertical profiles of the multicopter (~4 min). The SODAR had a temporal
resolution of 10 min, so only one value was available at each height. Wind direction and speed of the UAV data were in
good agreement with the recordings of the different instruments. During the flights at 09:01 UTC and 09:31 UTC, wind
direction was mainly from north to east with an increasing wind speed over time. For the first flight, spatial and temporal
averages of multicopter, SODAR and EC station were in agreement within 20–30° and a standard deviation of about ±20°
for wind direction. LIDAR data showed higher variability than other measurements but above 100 m data were in the same
range. Wind speed for all instruments was low with an average of about 1–1.5 m s$^{-1}$ and a standard deviation of about
±0.6 m s$^{-1}$. For the second flight, the same was true for wind direction, but greater differences occurred for wind speed.
While the multicopter and SODAR recorded a mean speed of 2.6 m s$^{-1}$ and 2.5 m s$^{-1}$, respectively, LIDAR and EC station
had 1.7 m s$^{-1}$ and 1.4 m s$^{-1}$, respectively. At this point it has to be highlighted that the instruments were not located at the
same place (distance 100–570 m from multicopter, see Fig. 1) and that time resolution varied. Besides, during north-easterly
winds generation of turbulence is likely at the edge of the forest, which is to the east of the investigation area. Accordingly,
differences were explainable, especially at heights up to 50 m.

## 3.2 Methane and meteorological conditions

In the night between 21 and 22 July 2015, methane measurements were made with the multicopter starting about 15 minutes after sunset (19:05 UTC) and extending over seven hours (Fig. 7). For comparison of tower and multicopter results, the subsequent measurements are displayed with orange points for tower data in 10 m and multicopter data with green ones also for 10 m. Short-term variations in methane concentration were detected by both techniques, even with the same extent (around 22:00 UTC). There was only one major deviation shortly past midnight when the multicopter measured a value of 2.45 ppm compared to 2.2 ppm at the tower. This may be due to the distance of approx. 5 m between tower and UAV and a time difference of around 30 s between those measurements. Overall, the two data sets were significantly correlated with a spearman correlation coefficient of 0.96. Calculation of the RMSE led to ±0.063 ppm. Consequently, the measurements on the moving platform were as representative as those of the stationary tower installation.

Considering the vertical methane profiles up to 50 m a.g.l., gradients were detectable during stable atmospheric conditions after sunset (Fig. 8). Data are shown for six flights with one-hour intervals beginning at 19:32 UTC and ending at 00:32 UTC. According to the potential temperature profiles, a stable stratification of the atmosphere developed after sunset indicated by increasing potential temperature with height. Its difference reached 5–6 K between ground and 50 m.

Thus, this overall stable stratification led to the reduced vertical mixing, and methane sources in the surroundings caused a concentration rise of 0.3 ppm after sunset within six hours. The mean background concentration measured during this campaign was 1.9 ppm. The concentration increased at each height with time, while accumulation started from the ground. Vertical gradients were already visible right after sunset, were intensifying until the measurement at 22:32 UTC, weakening afterwards and then intensifying again at 00:32 UTC. This variability in varying gradients was in agreement with changing meteorological conditions. Mean concentrations averaged over all measurements at each level were 2.091 ppm (10 m), 2.049 ppm (25 m), and 1.976 ppm (50 m).

According to the continuous measurements at the tower, the $CH_4$ concentration increased close to the ground even before sunset. The strongest increase was seen at all heights between 21:32 and 22.32 UTC with 0.25 ppm at 10 m, 0.15 ppm at 25 m, and 0.06 ppm at 50 m. Afterwards (23:32 UTC), concentration decreased in 10 and 25 m and increased in 50 m, leading to almost the same concentration in all heights (approx. 2.07 ppm).

Variations in agreement with a stabilization of the NBL were observed from the vertical potential temperature profiles. The stability of the atmosphere increased especially between 25 and 50 m until 22:32 UTC, while $CH_4$ accumulated in the NBL. Below 25 m, the atmosphere was slightly stable to neutral. In the following hour, a destabilization in the lowest 50 m of the atmosphere was detected and afterwards stable conditions developed again. This destabilization occurred simultaneously to the mixing of methane at all heights followed by a reestablished methane gradient. The results indicated a developing surface layer up to 25 m a.g.l. where methane accumulated, but exchange with air above was not completely inhibited likely due to the fact that turbulence was not totally suppressed.

Wind in this night was mostly from west to north-west with low speed between 1–2 m s$^{-1}$ and up to 3 m s$^{-1}$ in 50 m (Fig. 9) and is shown for the same times as in Fig. 8. During the first two hours, wind direction was roughly the same with height showing a variability of about 50° (W to NW), while wind speed was about 2–3 m s$^{-1}$. Afterwards, wind speed was lower at 10 and 25 m. Mean wind direction stayed between west and north-west in 25 and 50 m, while in 10 m it changed from south (21:32 UTC) to west (22:32 UTC) and back to south and south-west (23:32 UTC). So, southern directions were accompanied by a methane decrease, lower wind speeds and higher potential temperature. In contrast to that, at 22:32 UTC wind speed was higher than 1 m s$^{-1}$ and potential temperature was 4–5 K lower than the hour before and after. During the last flight, wind direction changed back to north-west with high variability of about 100° at 10 m and 25 m, which was not seen in the second height before. This higher variability occurred mostly during low wind speeds of 1–1.5 m s$^{-1}$.

## 4 Discussion

The presented results of the multicopter-based approach showed that extending measurements from towers have advantages because measurement height and location is more flexible. Methane concentration measurements at 10 m height were in good agreement with those on the tower and could be conducted at different heights. Therefore, we conclude that the technique of using a tube with a multicopter could be also applicable for other inert trace gases and related research questions. In view of the good agreement of tower and UAV-based methane concentrations, plausible methane gradients were observed during stable atmospheric conditions although the multicopter does stir air with its propellers. Palomaki et al. (2017) demonstrated in an experiment that wind speed at 30 cm above the multicopter is 0.5 m s$^{-1}$ due to spinning rotors. According to Alvarado et al. (2017) this influence is negligible at a distance of 40–45 cm above the multicopter.

Methane concentration increases close to the ground were found below a nocturnal inversion. Using a tethered balloon instead of a multicopter, Choularton et al. (1995) detected a concentration drop of 0.05 to 0.075 ppm from the inversion layer to the layer above. This is in agreement with our multicopter measurements in 10 and 25 m a.g.l. being below 0.1 ppm in the first half of the night while a stable stratification occurred.

The vertical range of measurements was limited by the payload capacity of the multicopter and the lateral extent of the measurements was restricted by electricity availability for the methane analyzer. Using a tethered balloon, Denmead et al. (2000) pointed to the problem that it was difficult to adapt to varying NBL heights with fixed installed sampling lines. This shortcoming can be overcome with the multicopter because hovering heights can be easily changed in the flight plan. A limitation of our setup was that the vertical range of 50 m is usually not enough to cover the whole NBL height. To overcome this limitation, a multicopter with a higher payload would be necessary with the ability of carrying a longer tube. Apart from that, the vertical extension of meteorological measurements to the NBL height without the tube would be benefiting for interpretation, although no methane data would be available.

In addition, no influence of the tube on the tilt angle could be detected while hovering at 10, 25 and 50 m. A negligible influence of payload was also found by Neumann and Bartholomai (2015). To each height, the multicopter had to lift more

weight, but the autopilot compensated this with the spinning speed of the propellers, which was significantly higher on the side where the tube was mounted. Therefore, it is recommended to mount the tube in the center for a better flight performance. Besides, non-gusty wind conditions are favorable to reduce the wind load on the tube.

The wind estimation carried out during hovering showed good agreement with the tower with a RMSE of 14.5° and 0.7 m s$^{-1}$ for wind direction and speed, respectively. These values were determined using a moving average of 10 s. Applying a 20 s moving average values of 12.5° and 0.6 m s$^{-1}$ are similar to those obtained by Neumann and Bartholomai (2015) for hovering. The advantage of our approach is that no wind tunnel experiments are necessary and that the experimental flights are easy to reproduce. Since the estimated errors were a result of only a 5 min flight, further experiments and comparisons would be necessary to confirm these values. Our experimentally determined relationship between TAS and the tilt angle is only valid for this hexacopter configuration and up to a speed of 6 m s$^{-1}$.

Although the multicopter-based wind estimation was biased, measurements show similar results and the results of the other instruments showed differences too. Wind speed differed up to about 1 m s$^{-1}$ and direction up to 50° above 50 m. Below this height, influences of topography, land use and horizontal distance as well as averaging time were more pronounced and differences larger. Horizontal distance to the multicopter was 370 m for LIDAR and 540 m for SODAR, while they had averaging times of 1 min and 10 min, respectively, compared to the 10 s moving average of the multicopter. Lothon et al. (2014), for example, found similar biased differences dependent on horizontal distance and land use during the BLLAST campaign. In addition, low wind speeds ($< 1$ m s$^{-1}$) lead to higher variability in wind direction as seen for LIDAR data.

This is because then the wind is not well coupled to the meso-scale flow, which is often leading to variable wind directions (Anfossi et al., 2005; Mahrt, 2010). The same is true for multicopter-based wind direction at 10 m during the nighttime flights, which mainly occurred during wind speeds of less than 2 m s$^{-1}$. With regard to wind estimation from horizontal flights, this is especially important because flying with a specific speed requires a certain tilt angle. If this angle is significantly larger than the wind induced angle, determination of wind contribution to the angle could be more difficult depending on the accuracy of measuring the angle.

Hovering close to the ground led to limitations in the estimation of wind from the flight control sensors. The propeller's downwash caused motion of air beneath the multicopter. These were compensated by changing the tilt angle, but did not reflect actual wind conditions below a height of 5–6 m a.g.l. The effect was stronger during calm conditions because the jet of perturbed air did not advect away effectively. For the same reason, the data collected during descent were not used to estimate wind conditions because the multicopter moved through its own downwash.

Since the thermocouple was placed below a rotor, discontinuities were found while hovering; the temperature measurement is rather representative for the volume around the multicopter than for a point. But this ensured a continuous flow around the sensor, which increased its response time. For analysis, temperature was averaged for hovering at each level during the methane measurements.

The combination of the wind and concentration measurements suggest that the significant methane increase between 21:32 UTC and 22:32 UTC was caused by emissions from the dairy farms (about 150–200 dairy cows) to the west of the

measurement location (about 600 m distance). Actually, the methane mixing ratio started to increase around 22:00 UTC (Fig. 7), when wind direction changed from more southern to predominating western directions (250–300°) with wind speeds of around 1.5 m s$^{-1}$ (Fig. 9). Below 25 m, the atmosphere was mixed according to the vertical potential temperature profile. Taking into account these conditions, dispersion of a methane plume is low. According to Dämmgen et al. (2012), an

emission rate of 14.5 g h$^{-1}$ cow$^{-1}$ can be assumed. This value was estimated for dairy cows in Bavaria (Germany) based on the IPCC guidelines (2006). Depending on the width of the methane plume(s) (100–500 m) coming from the farms, the methane concentration increase of about 0.15 ppm in half an hour would lead to emissions from about 90–450 cows. In comparison to the actual number of dairy cows measured methane concentrations were plausible. For further investigation, an approach similar to that of Hacker et al. (2016) would be suitable to calculate emission rates by flying upwind and

downwind of the farms and measuring the vertical and horizontal extent of the plume.

**5 Conclusion and outlook**

This case study demonstrated the feasibility of a multicopter-based approach to detach measurements of constituent mixing ratios and meteorological variables from fixed towers to achieve mobile and flexible investigations. Especially for regions difficult to access, sensible ecosystems or locations where high towers are prohibited, multicopter-based measurements could

be a suitable alternative. In addition, the results highlighted the need of both meteorological and methane measurements simultaneously. Information about potential temperature is important to determine the (in-)stability of the atmosphere and hence infer dispersion and mixing processes. Wind speed and direction provide information about the footprint, i.e. where the enhanced concentration originates. This is not only true for methane but is transferable to investigations of other trace gases and aerosols in the air. However, to apply budget methods for ground flux estimations as discussed by Denmead et al.

(2000), the vertical coverage needs to be extended, for example by utilization of a lightweight onboard methane sensor.
Also for horizontal methane investigations, a lightweight and small methane sensor onboard a multicopter would be beneficial. With such a sensor it would become possible to investigate the size of the methane plumes horizontally and vertically and determine methane fluxes. Besides, the investigation of further methane sources and sinks as well as their strengths is planned in that area. To this end, horizontal wind estimation is necessary.

**Competing interests**

The authors declare that they have no conflict of interest.

**Acknowledgments**

This research was supported by the TERrestrial Environmental Observatory (TERENO) pre-Alpine infrastructure funded by the Helmholtz Association and the Federal Ministry of Education and Research as well as the Ground Truth Demo and Test Facilities (ACROSS) infrastructure funded by the Helmholtz Association. The corresponding author was partly supported by a scholarship of the GRAduate School of Climate and Environment (GRACE) of the Karlsruhe Institute of Technology (KIT). Their support is highly acknowledged. We also thank the Scientific Team of the ScaleX Campaign 2015 for their contribution.

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

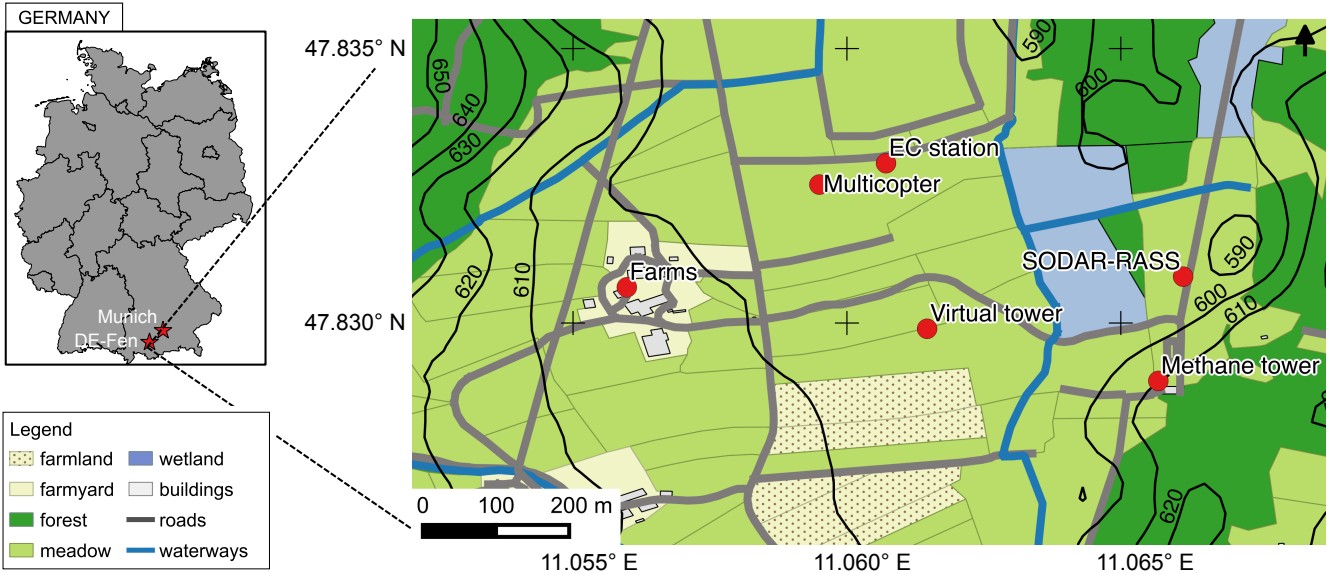

**Figure 1: Measurement site DE-Fen, Germany, with land use and ground-based instrumentation important for this study during the ScaleX campaign 2015. Contour lines stand for altitude (m) above sea level (QGIS, OpenStreetMap).**

| | |
|---|---|
| **1** | Air temperature and humidity sensors |
| **2** | Teflon tube |
| **3** | Tube extension above hexacopter |

**Figure 2: DJI F550 multicopter with installed sensors and tube.**

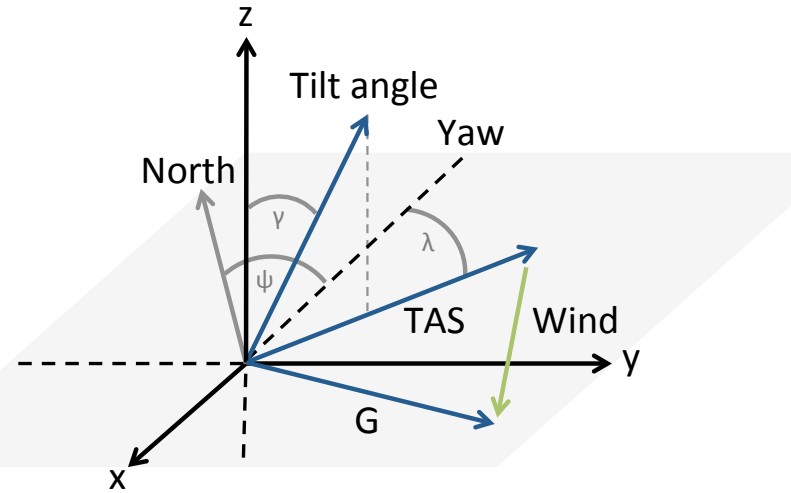

**Figure 3: Relationship between the tilt angle γ of the multicopter and the wind triangle with true air speed (TAS) vector, ground (G) vector and wind vector. Pitch angle is in x-axis and roll angle in y-axis direction. Yaw (ψ) is the viewing direction of the multicopter relative to north and the angle between TAS and yaw is λ.**

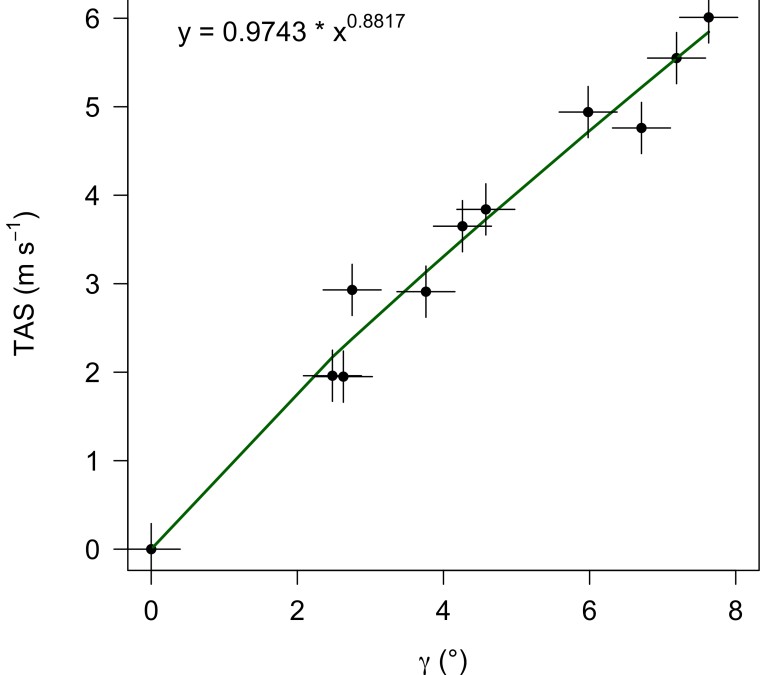

$$y = 0.9743 * x^{0.8817}$$

**Figure 4: Regression function of relationship between true air speed (TAS) and tilt angle ($\gamma$) experimentally determined with racetrack flights during calm wind conditions. The green line represents the fitted regression function and the error bars indicate the standard deviation of ±0.4° for the tilt angle and ±0.3 m s$^{-1}$, respectively.**

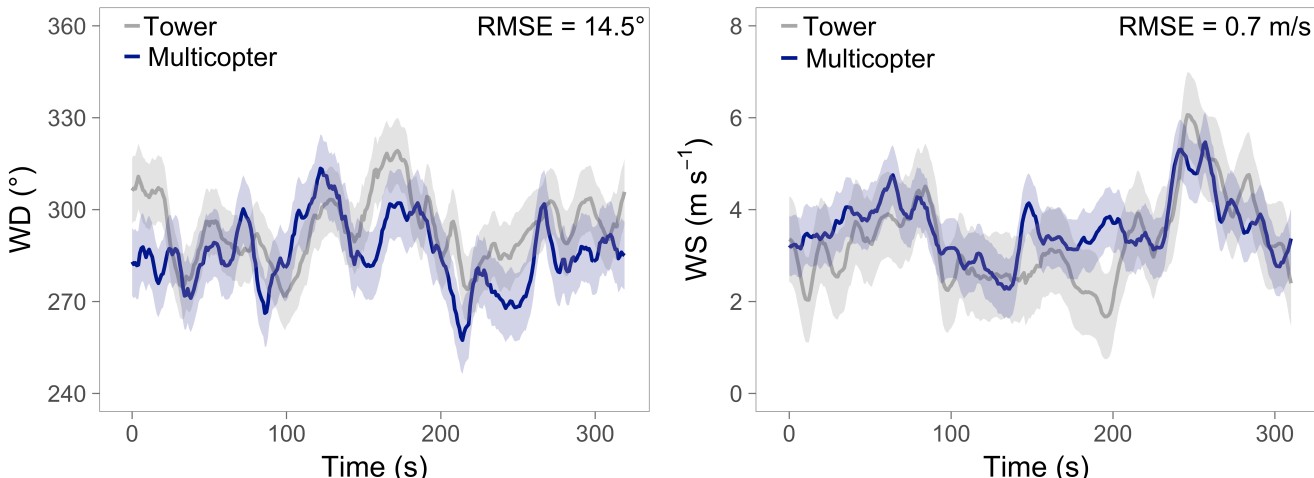

**Figure 5: Wind direction (WD) and speed (WS) comparison between tower (grey) and multicopter (blue) at 9 m a.g.l. over 5 min. The colored bands around the lines represent the standard deviation of each time series.**

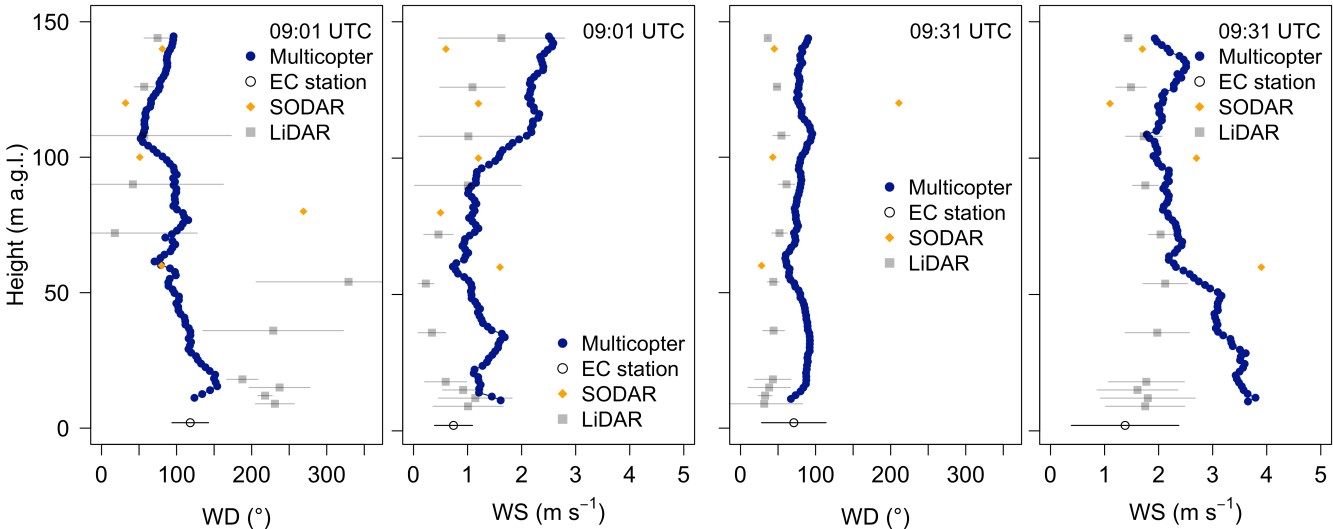

**Figure 6: Wind direction and speed profiles during two different flights: 09:01 UTC (left panels) and 09:31 UTC (right panels) on 15 July 2015. The blue profiles show multicopter data, dark grey circles represent EC station data, light grey squares LIDAR data and orange squares SODAR data. LIDAR and EC station data were averaged over the time the multicopter needed for the profile. Error bars show their standard deviation.**

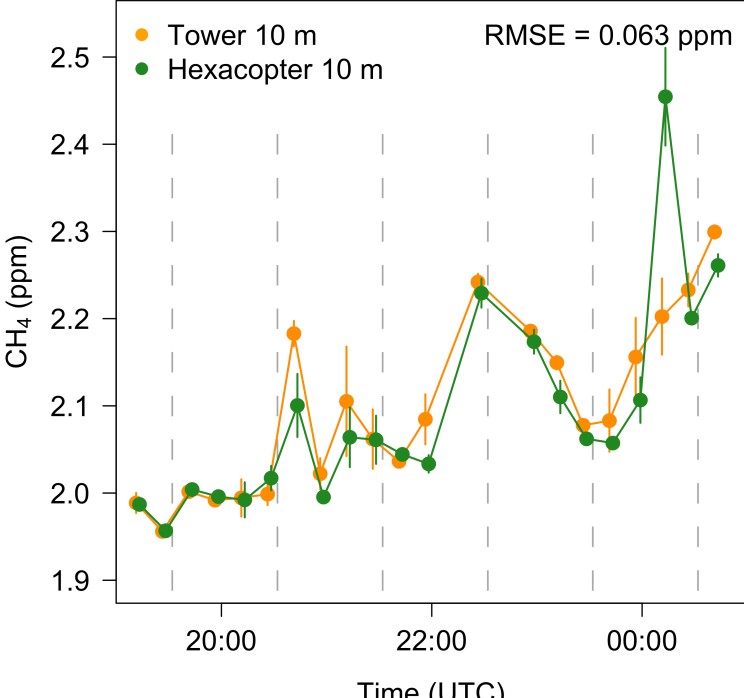

**Figure 7: Methane mixing ratio measured at the tower and with the multicopter in the night between 21 and 22 July 2015. Tower data were measured just before the 10 m data from the multicopter. Error bars show the standard deviation for each measurement averaged over 60 s. A standard deviation of 0.01 ppm or less cannot be shown because the size of the data point exceeds the error bar. The dashed grey lines represent the time for which vertical profiles are shown in Fig. 8 and 9. Local time (CEST) is UTC+2.**

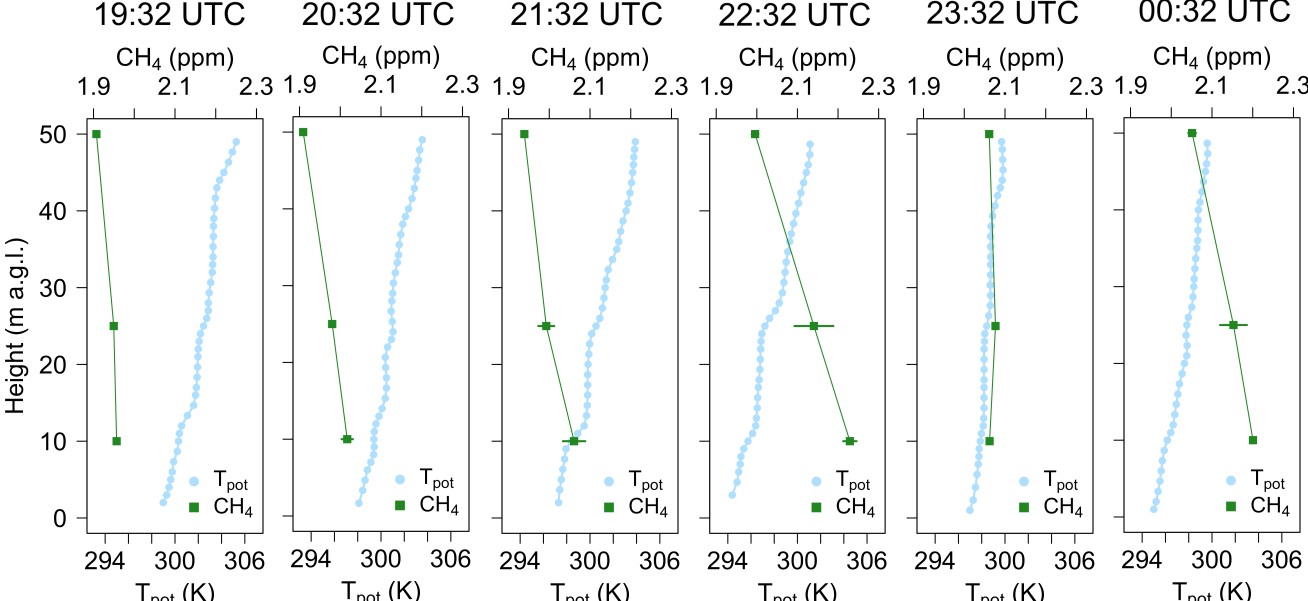

**Figure 8:** Vertical potential temperature ($T_{pot}$) profiles in blue and methane concentrations in green over six hours from 19:32 UTC (left) to 00:32 UTC (right) in the night 21 to 22 July 2015. This corresponds to: 21:32 CEST to 02:32 CEST (UTC+2). Air temperature was measured with the thermocouple (ascent data only) and $T_{pot}$ was calculated with the onboard pressure data from the autopilot. $T_{pot}$ was averaged at hovering levels and smoothed with a moving average (3 s). Error bars of methane concentration show the standard deviation for each measurement averaged over 60 s. A standard deviation of 0.01 ppm or less cannot be shown because the size of the data point exceeds the error bar.

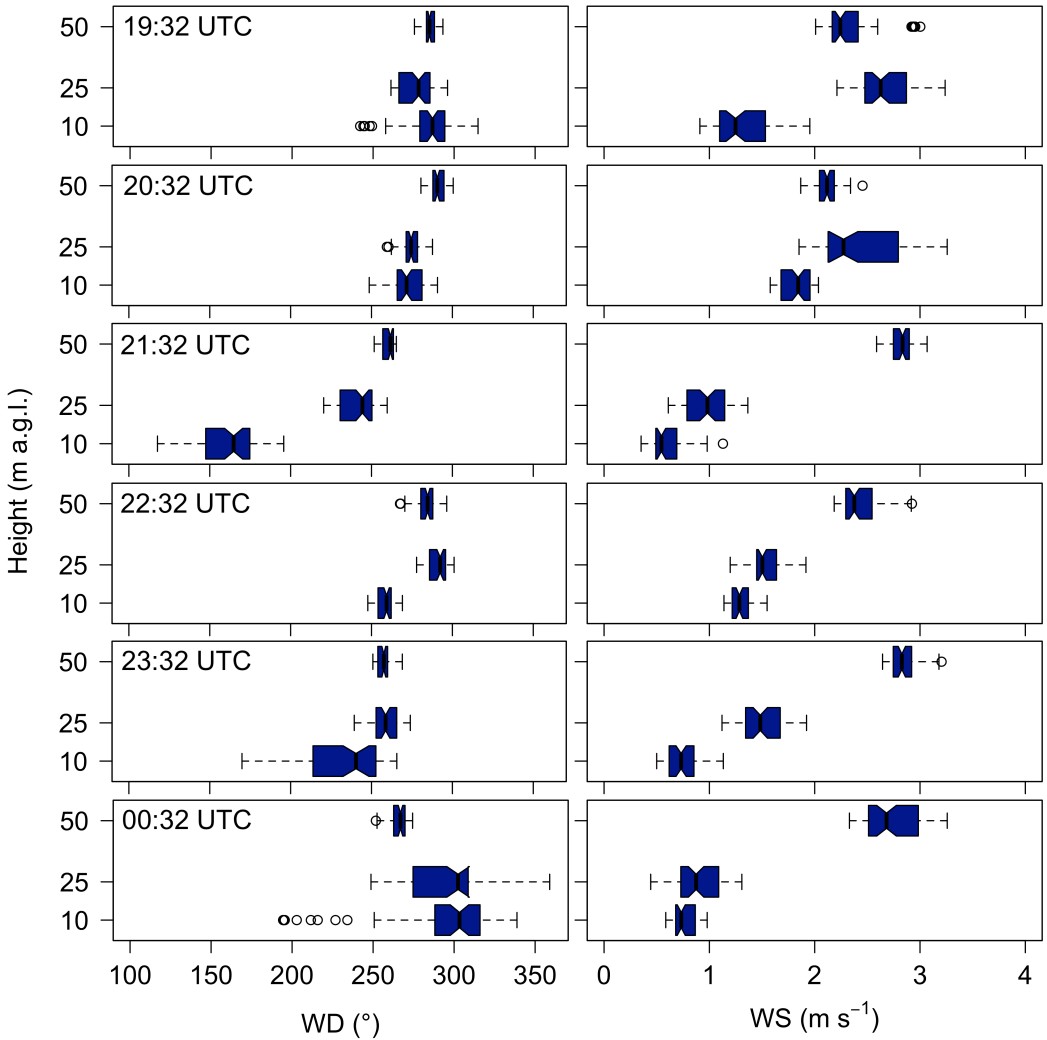

**Figure 9: Variability of wind direction (left) and speed (right) during 60 s hovering at 10, 25 and 50 m a.g.l. for flights between 19:32 UTC and 00:32 UTC in the night 21 to 22 July 2015. The blue box contains 50 % of the data and represents the interquartile range with the median as a black line. The dashed lines show maximum and minimum values in case those values are within the 1.5 interquartile range. Values outside this range (outliers) are represented with circles.**