# Peer review of "Simultaneous multicopter-based air sampling and sensing of meteorological variables"

_Atmospheric Measurement Techniques, 2017_

## Referee Comment (RC2) · J. Bange (Referee) · 18 Apr 2017

The manuscripts describes first results of experiments using a multicopter UAV for wind and methane measurements. For the wind vector, only attitude and GPS data were combined in a very simple flight mechanical model of the multicopter. This approach is not new, but only few publications exist so far. In order to avoid the usual problem with small multicopter regarding very limited payload, the authors use a ground-based gas spectrometer connected by a tube to the multicopter. This allows for vertical profiles of gas data in the lowest part of the atmospheric boundary layer / surface layer / Prandtl layer.

The results are in good agreement with other measurements (EC station, tower, lidar, sodar). This analysis is a valuable contribution to measurement technology in the

surface layer, although turbulence data cannot be achieved by the presented methods. The manuscripts meets the focus of the AMT journal perfectly.

Here are my comments:

a) general: although I am not a native speaker, I think the correct word before naming an altitude is 'at', not 'in' or 'for', e.g. text below Fig. 5, lines 14 and 27 on page 7, etc.

b) Page 5 and following: The procedure to find the relationship between tilt gamma and true airspeed TAS is based on the following assumptions:

1) TAS equals ground speed (measured using GPS) during absent wind (very calm wind, below 1 m/s)

2) TAS and gamma have a linear relation. Thus knowing gamma from attitude measurements leads directly to the TAS.

3) Since the difference between TAS and ground speed equals the wind vector, knowing the attitude / Euler angles allows the calculation of the wind vector (or at least an estimation)

First question addresses assumption #1: what is the mistake done to the TAS-gamma relationship by assuming zero wind during calm wind (1 m/s is not zero)?

Assumption #2: TAS and tilt angle gamma are not in a linear relation, but due to

Seddon, J. M., and S. Newman, 2011: Basic helicopter aerodynamics. 3rd ed., Wiley, 286 pp.,

and

Palomaki et al., 2017, Wind estimation in the lower atmosphere using multi-rotor aircraft, JTECH online: http://journals.ametsoc.org/doi/10.1175/JTECH-D-16-0177.1

(btw this article should be cited anyway),

 $TAS^2 = C * tan gamma$

Even assuming very small gamma angles, a Taylor series expansion would lead to

TAS2 approx C \* gamma, and not TAS approx C \* gamma !

This explains why the curve in Fig. 4 is not a straight line.

c) How was the wind direction estimated for situations with significant wind speed? Is the simple linear (or squared, see comment b) approach still valid for significant wind speed?

d) While the relation between gamma and TAS in Fig. 4 was found for calm wind situations only, the corresponding calibration experiment (race-tracks flights) was performed without the 70 m tube that provides the methane measurements in the following. The tube adds weight and moment of inertia to the multicopter and thus changes the flight mechanics. What / how large is the influence of the tube on the relation between gamma and TAS, and finally on the wind-vector estimation? I see that this aspect is addressed in line 4 on page 9 - but there it is just a statement, not explained or proven. Of course the autopilot could handle the extra load, but this does not mean that the gamma-TAS relation remains untouched.

\* Chapter 3.2: How much time did the multicopter spend at the three probing altitudes 10, 25 and 50 m?

\* Chapter 3.2 / Fig. 8: The curvature of the blue temperature lines in Fig. 8 is misleading, because it does not represent the vertical temperature profile of atmosphere, but was most likely caused by (all together)

1) non-stationarity of the ABL

2) the change of the multicopter flight mechanics before climbing to the next probing level (thrust) and thus the change of the wind field around the aircraft

3) sensor inertia of the quite slow thermocouple

I suggest to use averaged temperature data at the three probing levels only, similar to the CH4 data (green)

-

e) line 29 on page 7: How is the 'mean concentration of a gradient' defined?

f) line 30 on page 7: 'concentrations increased even before sunset' - how can you know? Because line 13 same page: 'starting 15 minutes after sunset'

g) line 7 on page 8: 'due to the fact that turbulence was not totally suppressed' - Well, this is more a guess rather than a fact, since turbulence was not measured.

h) line 22ff on page 9: You could visualise the mulitcopter downwash and quantify the downwash area using smoke. We did this - really easy to do and impressive.

i) section 3.2 and line 27ff on page 9: the methane data interpretation depends strongly on the accuracy of the methane concentration measurement, which is not addressed in the article. How accurate is the CRD spectrometer? See also missing error bars in Fig. 8 for additional statistical uncertainty.

j) Fig. 6 and line 13ff on page 9: it seems that the lidar wind direction at 9:00 UTC was corrupt due to very low wind speed. Same for all lidar data below 25 m. I looks like the lidar did not deliver reliable data at all under these conditions, and that this has nothing to do with horizontal separation from the sodar etc. This should be mentioned.

k) Fig. 7: for most data points the error bars are missing (since data points are a result of averaging it should be easy to add error bars)

I) Fig. 8: Since the data points (at least the CH4 concentration) are a result of averaging it would be easy to add error bars. This would give better confidence, or rather would help to see the significance of the concentration gradient described in the text, respectively.

m) Fig. 9: How were the errors calculated? What do the small circles represent?

---

## Referee Comment (RC3) · Anonymous Referee #3 · 24 May 2017

This manuscript describes a novel approach to measuring methane profiles in the surface layer using a ground-based instrument, a long ~70m sampling tube, and an unmanned aerial vehicle (UAV). The UAV was also equipped with meteorological sensors to measure temperature and humidity, and wind measurements are calculated based on multicopter in-flight data (pitch, roll, and yaw). With a few exceptions, the methane and wind measurements are in good agreement with EC station, tower, lidar and sonar measurements. The manuscript is generally well-written and should be published in AMT, once the following comments, in addition to those of the first two reviewers, have been fully addressed.

First, based on the placement of the temperature and humidity sensors (on the arm of the multicoptor below one rotor; Fig. 1), it is very likely that these meteorological measurements were negatively impacted by rotor-wash. Indeed, the discontinuities

in the potential temperature profiles at UAV sampling locations (Fig. 8) support this idea. I suggest placing the meteorological sensors closer to the methane inlet and away from the rotors. At the very least, this flaw in the method should be openly discussed and addressed in future studies. The authors should also comment on why the humidity data obtained from this sensor was not presented. Ideally, the authors could demonstrate using laboratory tests that the current flying geometry and sampling strategies do not adversely affect the either the methane measurements or the wind estimates.

The interpretation of the methane concentration gradients rely heavily on the interpretation of the meteorological conditions and changes in the surface layer with time. As a result, it would be helpful for figures 7 and 8 to show the local time, as well as UTC time.

The discussion on L25-30 is difficult to follow and should be re-written.

Finally, as noted in the manuscript, a more powerful UAV with a larger payload would enable longer profiles by ground-based gas spectrometer. Given a UAV with a larger payload, could the authors comment on what is the maximum altitude that could be reasonably sampled using this method, either due to prolonged residence time in the tubing, flow restrictions or other logistical concerns?

---

## Author Comment (AC1) · 20 Jun 2017

**Response to comments of Reviewers**

We would like to thank all the Reviewers for their thoughtful and valuable comments, which helped us to improve the content and quality of our manuscript. We addressed all comments and included changes in the manuscript as follows:

Blue: Comments of Reviewer

Black: Answers of Authors

*Black, italic, quotation marks: "Changes in the manuscript"*

Page and line references in our answers refer to the reviewed manuscript.

Additional references and updated Figures with captions were inserted at the end of this revision after addressing the comments of the three Reviewers.

**Reviewer #1**

Specific Comments:

**a) p. 1 line 11:** Define high spatial and temporal resolution. What is the resolution exactly?

**Answer:** We included a definition for high spatial and temporal resolution and changed the text to:

*"Vertical profiles of atmospheric variables in the SL at high spatial (meters) and temporal (1 Hz and better) resolution increase our understanding of these interactions, but are still challenging to measure appropriately."*

**b) p. 1 line 13:** Scanning lidar observations (of various kinds: Doppler, DIAL, Raman) can measure winds, temperature, moisture, and some trace gas quantities (such as ozone) with high-vertical and spatial resolution (on the order of meters to 10s of meters) at low elevations angles and over large horizontal areas close to the ground. Thus, the broad statement that 'remote sensing techniques ... are challenged to achieve sufficient detail near the ground' should be modified.

**Answer:** We agree with this and changed the sentence to:

*"At the same time, most remote sensing techniques and aircraft measurements have limitations to achieve sufficient detail close to the ground (up to about 50 m)."*

**c) p. 1 line 14:** Change 'horizontal sounding' to 'horizontal transect'. A 'sounding' implies a vertical profile by definition.

**Answer:** We changed the text to:

*"Vertical and horizontal transects of the PBL can be complemented by unmanned aerial*

*vehicles (UAV)."*

**d) p. 1 line 24:** Clarify that the UAV measures a continuous profile, in comparison that towers only measure at discrete levels where instrumentation are installed.

**Answer:** Since in this context it was pointed out that the vertical height of tower measurements was extended by UAV measurements, we think that at this point an explanation is not necessary. In addition, this is too much information for the abstract. Instead, we stated on p. 2, l. 3:

*"The operation of towers is fixed to a certain location and the vertical information is limited to the height of the tower as well as to discrete levels at the tower."*

**e) p. 2 line 9:** Again, there have been studies using scanning lidar to make profiles where the minimum height above ground is 10-20 m (see Langford et al., 2015 (ozone profiles), Banta et al., 2013 (wind profiles), Yabuki et al. (water vapor), and Hammann et al., 2015 (temperature)). Data shown in these studies contradicts the statement that ground-based remote sensing cannot measure near the surface.

**Answer:** Looking at the literature you provided, it is indeed possible to measure at low altitudes with a LIDAR. But considering for example acoustic instruments, this is not the case. Therefore, we differentiated LIDAR and acoustic methods and changed the sentence. References were included, too:

*"Considering ground-based remote sensing methods, data of vertical profiles from low altitudes up to about 50 m above ground level (a.g.l.) are hardly usable (e.g. acoustic instruments), but possible with LIDARs applying certain scan patterns with low elevation angles at the position of such an instrument (Emeis, et al., 2009; Banta et al., 2013; Korhonen et al., 2014; Hammann et al., 2015).*

**f) p. 2 lines 10-14:** This paragraph seems out of place here. I recommend it gets moved and integrated into the paragraph at line 33 or before that paragraph.

**Answer:** Yes, we agree with the reviewer that this paragraph fits better before the last paragraph of the introduction and we moved it there.

**g) p. 2 lines 16-32:** Most of the cited works here are within the past 2 years, however UAVs for atmospheric research have been more widely used for ~10 years. I recommend that the authors provide more details on the pioneering (first) uses of UASs for atmospheric measurements.

**Answer:** Yes, it is true that UAVs were already used for some decades. We inserted the following paragraph to show the pioneering usage of UAVs and included the citations in the

references:

*"From the 1970s on, UAVs were used for atmospheric research, for example for convective processes (Konrad et al., 1970; Rennó and Williams, 1995) and weather forecast (Holland et al., 1992; McGeer and Holland, 1993), as well as for vertical sounding of the planetary boundary layer (Egger et al., 2002; Soddell et al., 2004; Spiess et al., 2007)."*

**h) p. 3 line 20:** Add a statement 'location of instruments is shown in Fig. 1'.

**Answer:** We included the following sentence:

*"An overview about the location of instruments is given in Fig. 1."*

**i) p. 3 line 24 & 27:** Remove references to Fig. 1, seems odd here since the figure does not show the resolution. Seems like odd placements.

**Answer:** We removed the references to Fig. 1 in those two lines.

**j) p. 4 line 14:** Here, the authors discuss night flights. In the next paragraph, it states that the aircraft has permission to fly in the daytime. Did the authors fly at night when there was no permission? Alternatively, if they did get special permission for the night flights, this should be explained so that it is clear the flights were legally conducted.

**Answer:** We agree that this statement is misleading. We exchanged *"daytime"* with *"time of the day"* which makes it clear that day and night is meant.

**k) p. 4 line 30:** What are the specifications for the pressure sensor? This is important to identify the accuracy of the potential temperature calculation.

**Answer:** We included the specification of the pressure sensor as follows:

*"The used pressure sensor is a MS5611-01BA03 (AMSYS, Mainz, Germany) and is able to resolve an altitude of 10 cm corresponding to a precision of about ±0.02 hPa."*

**l) p. 5 line 12:** Can this method be used to retrieve 'w' as well as u/v? If not, specify that the horizontal wind speed is measured.

**Answer:** No, the vertical wind component cannot be estimated with that method, only the horizontal components. We specified the text:

*"Consequently, in the easiest case the direction of TAS represents the horizontal wind direction and the length of the TAS vector the horizontal wind speed."*

**m) p. 5 line 30 / Fig. 4:** Was the RMSE of the wind speed measurement a function of speed? Specifically, were high or low wind speeds measured more accurately? In Fig. 4, it may be more useful to show errorbars for each data point separately, if the statistics are

robust enough to calculate the RMSE at each speed bin. Similarly, was the RMSE for the tilt angle constantly 0.4 deg at all speeds?

**Answer:** The wind speed estimation is a function of TAS, which is equal to the flight speed under completely calm conditions (no wind). Since the experiment took place outside, this was not totally true (wind speed < 1 m s$^{-1}$). Therefore, a specific flight speed can be assigned to a specific tilt angle, which shows a variability depending on actual atmospheric conditions. The less small-scale turbulence was present the lower the variability of the tilt angle. But this variability was independent on the flight speed. So within the measurement accuracy, it is not possible to determine whether the estimation of low wind speeds is more accurate than higher wind speeds or the other way round. However, the relative error of wind speed estimation is higher for lower than for higher wind speeds, considering a mean TAS error.

Tow sentences were added in the manuscript, the first on p. 5, l. 27 and the second on p. 5, l. 31:

*"While the ground speed was kept constant by the GPS (< ±0.2 m s$^{-1}$), the variability of the assigned tilt angle was dependent on atmospheric conditions."*

*"This mean error of TAS leads to a higher relative error for low wind speeds than for higher wind speeds."*

**Answer:** In order to give numbers for windy conditions, we added this to the sentence:

*"During windy conditions (3–5 m s$^{-1}$) the multicopter was hovering for 5 min close to the tower at a distance of approximately 5 m."*

**o) p. 6 line 15:** Since the inlet for the methane measurement was only 30 cm above the propellers, how was it ensured that the measurement was not disturbed by the flow above the spinning propellers? The methane measurement was taken as a 60 s average; during that 60 s, surely the measurement was distorted as air is transported downward by the propellers.

**Answer:** Yes, it is true that during 60 s the air was distorted due to the spinning propellers. Simulations and experiments showed that the air above a multicopter is affected up to 2 m (Haas et al., 2014), but most of the influence is within the first 0.5 m (Alvarado et al., 2017). Unfortunately, Haas et al. (2014) belongs to the grey literature and is not cited in the manuscript.

A sentence was included in the manuscript on p. 6, l.18 and literature was added to the references:

*"Alvarado et al. (2017) experimentally determined a distance of 40–45 cm above the multicopter, where the influence of the rotors to air speed decreases significantly. So, the*

*methane mixing ratio is actually not a point measurement but valid for a volume."*

Reference Haas et al. (2014):

Haas, P. Y., Balistreri, C., Pontelandolfo, P., Triscone, G., Pekoz, H., and Pignatiello, A.: Development of an unmanned aerial vehicle UAV for air quality measurement in urban areas, In 32nd AIAA Applied Aerodynamics Conference, 2014.

**p) p. 6 line 26:** Looking at Fig. 5 (and the standard deviations given on lines 23-24), it is clear that the multicopter does not capture the full range of variability that the sonic anemometer measures. The extrema measured by the tower are much larger (1.5 m/s and 6 m/s) compared to the hexacopter. Presumably the hexacopter measurement is some kind of an average, possibly because the volume the hexacopter takes up is much larger than the sonic measurement volume and the hexacopter has inertia. Have you tried running a smoothing filter (or low-pass filter) over the sonic data to quantify these effects? These effects should be noted in the manuscript, as the wind speed deviation should not be used as a measure of turbulence (since it does not compare favorably, at least here, to the sonic anemometer measurement).

**Answer:** Yes, the multicopter does not capture the full range of variability compared to the anemometer because of inertia. Actually, we used the same 10 s moving average for the anemometer as for the multicopter. To make that clearer, we added this to the sentence on p. 6, line 25:

*"For both time series the 10 s moving average was applied resulting in a RMSE between multicopter and tower of 14.5° and 0.7 m s$^{-1}$, respectively."*

Additionally, we included following sentences at the end of this paragraph to note the effects of inertia:

*"Since the volume of the multicopter is larger compared to the measurement path of the sonic anemometer, the multicopter does not react to the small turbulent elements, the so-called eddies, and therefore cannot capture the full range of wind speed. In addition, the multicopter has inertia due to its weight. Consequently, the wind speed deviations measured by the multicopter should not be used as information about atmospheric turbulence."*

In order to see the standard deviation of multicopter and tower measurements in the plot (Fig. 5) and not only have the numbers in the text, we added colored bands to the time series lines. Accordingly, an explanation is given in the caption:

*"The colored bands around the lines represent the standard deviation of each time series."*

**q) Fig. 6:** To increase readability and for ease of comparison, I suggest averaging the lidar and EC data over the profiles. To convey the variability, error bars could be used.

**Answer:** Indeed, averaging the data makes it easier to understand and the variability is still visible with the error bars. Additionally, we extended the caption of Fig. 6 to:

*"LIDAR and EC station data were averaged over the time the multicopter needed for the profile. Error bars show their standard deviation."*

**r) p. 7 line 8:** Could topography cause the observed wind speeds at the sodar site and multicopter site to be higher? Without a map of the topography, this is difficult to determine.

**Answer:** In order to have an impression of topography from the investigation site, we included contour lines in Fig. 1. Generally, topography as well as different land uses can cause different wind conditions comparing the available instruments. While topography is the predominating reason for west wind situations, during north-east wind, the edge of the forest causes the generation of turbulence leading to spatial differences in wind conditions. Therefore, we included the following sentence (p. 7, l. 9):

*"Besides, during north-easterly winds generation of turbulence is likely at the edge of the forest, which is to the east of the investigation area."*

**s) p. 7 line 22 and Fig. 7:** When was sunset? I suggest adding a vertical line on Fig. 7 denoting sunset time. Also, keep units consistent for CH4 (either ppm or ppb). For the caption of Fig. 7, clarify what the error bars show exactly (standard deviation of what, the 1-min timeseries)?

**Answer:** On that day, sunset was at 19:05 UTC at the investigation site. Since this was about 15 min before our UAV measurements started, the vertical line indicating sunset would be on top of the y-axis. So, we included the time of sunset on p.7, l. 13:

*"In the night between 21 and 22 July 2015, methane measurements were made with the multicopter starting about 15 minutes after sunset (19:05 UTC) and extending over seven hours (Fig. 7)."*

*"ppb"* was replaced by *"ppm"* throughout the manuscript: p. 1, l. 20; p. 7, l. 19, 26, 30; p. 8, l.25f; p. 10, l. 1

In the caption of Fig. 7 the following sentence was added:
*"Error bars show the standard deviation for each measurement averaged over 60 s."*

**t) Fig. 8:** i) I suggest changing this to a two panel plot, one panel each for the methane and one for the potential temperature with separate lines for different profile times. With having all

of the profiles on one plot, it will be much easier to see the change in the profile over time, even the small changes and increase in stabilization. ii) Can error bars be added for each measurement? It may clutter the plot too much, but it is something to consider. iii) It would be better to change potential temperature units to Kelvin, as it is usually presented. iv) Also, please explain in the text why there is a discontinuity at each height where the multicopter hovers. Is it due to the fact that the temperature is evolving over the time it is hovering, or some kind of hysteresis in the sensor?

**Answer:** i) We have looked into this but found that the current layout provides most clarity and least clutter. Since the x-axes have the same range for each time step, we think that changes can be seen easily, better than using six different colors and lines crossing each other.

ii) Error bars were added for methane in Fig. 8.

iii) Unit of potential temperature was changed to Kelvin.

iv) Temperature discontinuity was caused by the spinning of the rotors because they stir the air around the multicopter and not because of hysteresis of the sensor. Reviewer 2 suggested to average potential temperature at hovering levels as it is done for methane as well and we changed the profiles and caption of Fig. 8 accordingly to this suggestion:

"$T_{pot}$ was averaged at hovering levels and smoothed with a moving average (3 s)."

**u) p. 7 line 28:** How could the gradients be both intensifying and weakening over time? Please clarify.

**Answer:** With stabilization of the atmosphere a vertical $CH_4$ gradient developed, which was intensifying at first. Due to changing meteorological conditions, on the one hand air was mixed vertically due to a weakening of the stable stratification and on the other hand another air mass was advected due to wind direction change. This led to mixing of $CH_4$ too. Afterwards, meteorological conditions were similar to those before at the beginning of the night and the vertical $CH_4$ gradient could develop again.

For clarification we rewrote the sentence and gave a short explanation, which is addressed in more detail in the manuscript on p. 8, l. 1:

"Vertical gradients were already visible right after sunset, were intensifying until the measurement at 22:32 UTC, weakening afterwards and then intensifying again at 00:32 UTC. This variability in varying gradients was in agreement with changing meteorological conditions."

**v) p. 7 line 31:** There was an increase at 50 m as well, it simply was not as large of an increase.

**Answer:** It is true and we removed this sentence and wrote instead:

*"The strongest increase was seen at all heights between 21:32 and 22:32 UTC with 0.25 ppm at 10 m, 0.15 ppm at 25 m, and 0.06 ppm at 50 m."*

**w) p. 8 line 23:** With regard to the statement 'although the multicopter does stir air with its propellers', are there any tests you can do to quantify these effects on the inlet? Measure the flow disturbance at the inlet itself to infer any vertical transport during the 60 sec hovering periods?

**Answer:** Experiments regarding this issue were already done by Alvarado et al. (2017) and Palomaki et al. (2017). Palomaki et al. (2017) used the same hexacopter configuration except with larger propellers (9'' versus 14'') as we in our study leading to an air speed of 0.5 m s$^{-1}$ at 30 cm above the multicopter. Alvarado et al. (2017) found even less influence above 40–45 cm.

For additional information, this was inserted in the manuscript on p. 8, l. 23:

*"Palomaki et al. (2017) demonstrated in an experiment that wind speed at 30 cm above the hexacopter is 0.5 m s$^{-1}$ due to spinning rotors. According to Alvarado et al. (2017) this influence is negligible at a distance of 40–45 cm above the multicopter."*

**x) p. 9 line 8:** Given these values were made using only 5-min of data under a small range of values, the robustness of these statistics is questionable. I suggest adding a qualifying statement here to emphasize these limitations. Also, the RMSE of the wind direction is highly dependent on the wind speed. At low wind speeds (<1 m/s), the RMSE of the wind direction measurement would be much larger.

**Answer:** Yes, it is true that this is rather a qualifying result and not a quantitative one. Since the experiment already had the limitation of < 1 m s$^{-1}$, wind speeds below that cannot be determined.

*"Since the estimated errors were a result of only a 5 min flight, further experiments and comparisons would be necessary to confirm these values. Our experimentally determined relationship between TAS and the tilt angle is only valid for this hexacopter configuration and up to a speed of 6 m s$^{-1}$."*

**y) p. 9 line 13:** This statement should be modified. The vertical wind profiles were not in good agreement. The wind speed from the mulitcopter was systematically larger, and the wind direction was also biased high.

**Answer:** Taking into account the uncertainties of the wind measurements as well as topography, horizontal distance and averaging time, wind estimation is in good agreement from our point of view. We included the statement about the biased results and specified differences:

*"Although the multicopter-based wind estimation was biased, measurements show similar results and the results of the other instruments showed differences too. Wind speed differed up to about 1 m s$^{-1}$ and direction up to 50° above 50 m. Below this height, influences of topography, land use and horizontal distance as well as averaging time were more pronounced and differences larger. Horizontal distance to the multicopter was 370 m for LIDAR and 540 m for SODAR, while they had averaging times of 1 min and 10 min, respectively, compared to the 10 s moving average of the multicopter."*

**z) p. 9 line 16:** Were the differences that Lothon et al (2014) systematically different (biased), or were the differences more scattered (inaccurate)?

**Answer:** The differences Lothon et al. (2014) found were biased dependent on horizontal distance and land use. We change the sentence to:

*„Lothon et al. (2014), for example, found similar biased differences dependent on horizontal distance and land use during the BLLAST campaign."*

**aa) p. 9 line 23:** Should 'in the west' actually be 'to the southwest'?

**Answer:** Looking at Fig. 1, the "Farms" are to the west of the "Methane tower". To avoid ambiguities of the flight locations, we added a sentence at the end of section 2.4:

*"While most of the flights were done above the grassland site south of the EC station as shown in Fig. 1, the flights including methane measurements took place close to the methane tower in the south-east of the investigation area."*

Technical corrections:

**a) p. 3 line 2:** Should this be Sect. 2.2 (not 2.4)?

**Answer:** Sect. 2.2 was added, because this section explains the measurement device and in Sect. 2.4 the measurements themselves are explained.

**b) p. 3 line 9 and p. 4 line 29:** 'at' instead of 'with' 10 Hz.

**Answer:** Corrected as suggested.

**c) p. 4 line 22:** 'approximately' instead of 'approx.'

**Answer:** Changed as suggested.

**d) p. 4 line 23:** Use 'At 50 m length' instead of 'In 50 m height'.

**Answer:** Changed as suggested.

**e) p. 5 line 17:** 'simultaneous' instead of 'simultaneously'

**Answer:** Changed as suggested.

**f) p. 6 line 6:** 'a' instead of 'an'
**Answer:** Changed as suggested.

**g) p. 6 line 7:** 'at' instead of '@'
**Answer:** Changed as suggested.

**h) p. 7 line 29:** Remove 'of these gradients' and 'respectively'
**Answer:** Removed as suggested.

**i) p. 7 line 31:** Remove 'remarkably'
**Answer:** Removed as suggested.

**j) p. 9 line 20:** Change 'is' to 'if'
**Answer:** Corrected as suggested. In addition, "is" was inserted:
*"If this angle is significantly…"*

**k) p. 10 line 11:** Add missing word, 'hence infer dispersion and mixing processes'.
**Answer:** The word "infer" was inserted.

**Reviewer #2**

**a) general:** although I am not a native speaker, I think the correct word before naming an altitude is 'at', not 'in' or 'for', e.g. text below Fig. 5, lines 14 and 27 on page 7, etc.
**Answer:** According to your suggestion we changed it in the caption of Fig.5, on page 7 lines 10,14,27,33, page 8 lines 5,10,15 and page 9, line 17.

**b) Page 5 and following:** The procedure to find the relationship between tilt gamma and true airspeed TAS is based on the following assumptions:
1) TAS equals ground speed (measured using GPS) during absent wind (very calm wind, below 1 m/s)
2) TAS and gamma have a linear relation. Thus knowing gamma from attitude measurements leads directly to the TAS.
3) Since the difference between TAS and ground speed equals the wind vector, knowing the attitude / Euler angles allows the calculation of the wind vector (or at least an estimation)

First question addresses assumption #1: what is the mistake done to the TAS-gamma relationship by assuming zero wind during calm wind (1 m/s is not zero)?

**Answer:** Concerning assumption 1: The length of TAS equals the length of the ground vector, but the direction of both vectors is in opposite direction.

As written on p. 5, l. 33ff, the variability of the tilt angle is 0.7° ±0.3° during wind speeds < 1 m/s. Together with the mean error of gamma of ±0.4° in the regression, the maximum absolute error is 0.7° ±0.7° which corresponds to 0.7 m s$^{-1}$ ±0.6 m s$^{-1}$.

Assumption #2: TAS and tilt angle gamma are not in a linear relation, but due to

Seddon, J. M., and S. Newman, 2011: Basic helicopter aerodynamics. 3rd ed., Wiley, 286 pp.,

and

Palomaki et al., 2017, Wind estimation in the lower atmosphere using multi-rotor aircraft, JTECH online: http://journals.ametsoc.org/doi/10.1175/JTECH-D-16-0177.1 (btw this article should be cited anyway),

TAS^2 = C * tan gamma

Even assuming very small gamma angles, a Taylor series expansion would lead to

TAS^2 approx C * gamma, and not TAS approx C * gamma !

This explains why the curve in Fig. 4 is not a straight line.

**Answer:** It was not an assumption from us that TAS and gamma have a linear relationship. This was found experimentally with the racetrack flights. This method is widely used for aircraft measurements of the 3D wind vector with turbulence probes (e.g. multi-hole probe). The difference is that an aircraft is flying with a constant true air speed and a varying ground speed dependent on the wind conditions. To counteract these conditions, a lead angle is used. In contrast, a multicopter has a continuous ground speed and varying true air speed; the wind is compensated by changing the tilt angle.

Palomaki et al. (2017) was included in the manuscript (p. 2, l. 28) and references:

*"In addition, Neumann and Bartholomai (2015) and Palomaki et al. (2017) showed that the onboard flight control sensors can be used to derive wind estimates from a multicopter's attitude control data."*

c) How was the wind direction estimated for situations with significant wind speed? Is the simple linear (or squared, see comment b) approach still valid for significant wind speed?

**Answer:** Since the maximum flight speed of the multicopter is about 10 m s$^{-1}$, significant wind would be in the range of 7–8 m s$^{-1}$, which is 70–80 % of the speed. Our experiment to determine the regression function included that speed, but within the 120 m long straight legs

this speed could not be reached. Therefore, the approach is valid up to ~6 m/s and not for significant wind speed. Further tests for significant wind speed were not done yet and for the flights in this manuscript this was not necessary.

d) While the relation between gamma and TAS in Fig. 4 was found for calm wind situations only, the corresponding calibration experiment (race-tracks flights) was performed without the 70 m tube that provides the methane measurements in the following. The tube adds weight and moment of inertia to the multicopter and thus changes the flight mechanics. What / how large is the influence of the tube on the relation between gamma and TAS, and finally on the wind-vector estimation? I see that this aspect is addressed in line 4 on page 9 - but there it is just a statement, not explained or proven.
Of course the autopilot could handle the extra load, but this does not mean that the gamma-TAS relation remains untouched.
**Answer:** Concerning the effect of the tube to the wind estimation, Neumann and Bartholomai (2015) stated that the influence of the payload is negligible regarding the TAS-gamma relationship. Their tests included a payload of 27 % of the takeoff weight. In our case, the additional payload was 30 % at 50 m a.g.l. and was therefore in the same range. Additionally, since the wind data were only used during hovering, no change in payload occurred compared to ascending to the next level.
We included a statement in the manuscript (p.9, l. 4f):
*"A negligible influence of payload was also found by Neumann and Bartholomai (2015)."*

* Chapter 3.2: How much time did the multicopter spend at the three probing altitudes 10, 25 and 50 m?
**Answer:** The multicopter hovered 60 s at each level, which is written on p. 6, l. 15.

* Chapter 3.2 / Fig. 8: The curvature of the blue temperature lines in Fig. 8 is misleading, because it does not represent the vertical temperature profile of atmosphere, but was most likely caused by (all together)
1) non-stationarity of the ABL
2) the change of the multicopter flight mechanics before climbing to the next probing level (thrust) and thus the change of the wind field around the aircraft
3) sensor inertia of the quite slow thermocouple I suggest to use averaged temperature data at the three probing levels only, similar to the CH4 data (green)
**Answer:** We agree with statement 1) and 2), but with a time response better than 1 Hz the thermocouple is not slow and the discontinuities of potential temperature were rather caused by the rotors downwash and drawing air from above, which was mentioned by Reviewer #3

too.

But we agree with averaging the temperature data as for methane and smoothed the profile using a 3 s moving average. This was included in the caption of Fig. 8:

*"$T_{pot}$ was averaged at hovering levels and smoothed with a moving average (3 s)."*

**e) line 29 on page 7**: How is the 'mean concentration of a gradient' defined?

**Answer:** It means that the data at each height were averaged over all measurements. So the mean concentrations at each height are not the same and therefore there is a gradient. For clarification in the text, it was changed to:

*"Mean concentrations averaged over all measurements at each level were 2.091 ppm (10 m), 2.049 ppm (25 m), and 1.976 ppm (50 m)."*

**f) line 30 on page 7:** 'concentrations increased even before sunset' - how can you know? Because line 13 same page: 'starting 15 minutes after sunset'

**Answer:** Since methane measurements were done continuously at the tower, we know that close to the ground the concentration already increased before sunset. We included this explanation as follows:

*"According to the continuous measurements at the tower, the $CH_4$ concentration increased close to the ground even before sunset."*

**g) line 7 on page 8:** 'due to the fact that turbulence was not totally suppressed' - Well, this is more a guess rather than a fact, since turbulence was not measured.

**Answer:** Yes, it is true that turbulence was not measured. But since methane was mixed, vertical exchange was present. To make it clear that this is a guess we changed the sentence to:

*"The results indicated a developing surface layer up to 25 m a.g.l. where methane accumulated, but exchange with air above was not completely inhibited likely due to the fact that turbulence was not totally suppressed."*

**h) line 22ff on page 9:** You could visualise the mulitcopter downwash and quantify the downwash area using smoke. We did this - really easy to do and impressive.

**Answer:** Thank you for this suggestion, we will try it out.

**i) section 3.2 and line 27ff on page 9**: the methane data interpretation depends strongly on the accuracy of the methane concentration measurement, which is not addressed in the article. How accurate is the CRD spectrometer? See also missing error bars in Fig. 8 for additional statistical uncertainty.

**Answer:** Yes, it is right that there is no information about the accuracy of the CRD spectrometer. We included this information in section 2.1 where the instrument was introduced. Error bars were included in Fig. 8.

*"Methane mixing ratios were determined using a cavity ring down (CRD) spectrometer*

*(G2508, Picarro Inc., Santa Clara, CA, USA) with an accuracy of < 0.007 ppm."*

**j) Fig. 6 and line 13ff on page 9:** it seems that the lidar wind direction at 9:00 UTC was corrupt due to very low wind speed. Same for all lidar data below 25 m. I looks like the lidar did not deliver reliable data at all under these conditions, and that this has nothing to do with horizontal separation from the sodar etc. This should be mentioned.

**Answer:** Yes, the determination of wind direction is more difficult when wind speed is low. But this has nothing to do with height. At 9:00 UTC, wind speed measured by the LIDAR was even lower at around 50 m leading to higher variability in direction. At 9:30 UTC, wind speed was higher and wind direction variability was lower. For clarity, we included the following sentence at the end of this paragraph:

*"In addition, low wind speeds (< 1 m s$^{-1}$) lead to high variability in wind direction as seen for LIDAR data."*

Due to the insertion of this sentence, the following two sentences were swapped and modified as follows:

*"This is because then the wind is not well coupled to the meso-scale flow, which is often leading to variable wind directions (Anfossi et al., 2005, Mahrt, 2010). The same is true for multicopter-based wind direction at 10 m during the nighttime flights, which mainly occurred during wind speeds of less than 2 m s$^{-1}$."*

**k) Fig. 7:** for most data points the error bars are missing (since data points are a result of averaging it should be easy to add error bars)

**Answer:** In Fig. 7, the error bars are not missing, they are sometimes just not larger than the size of the data point. This includes a standard deviation of 10 ppb or less. Therefore, an explanation was inserted in the caption of Fig. 7:

*"Error bars show the standard deviation for each measurement averaged over 60 s. A standard deviation of 0.01 ppm or less cannot be shown because the size of the data point exceeds the error bar."*

**l) Fig. 8:** Since the data points (at least the CH4 concentration) are a result of averaging it would be easy to add error bars. This would give better confidence, or rather would help to see the significance of the concentration gradient described in the text, respectively.

**Answer:** Error bars were included in Fig. 8. As for Fig. 7, a standard deviation of 10 ppb or less cannot be represented and therefore error bars are not visible. We added two sentences in the caption of Fig. 8:

*"Error bars of methane concentration show the standard deviation for each measurement averaged over 60 s. A standard deviation of 0.01 ppm or less cannot be shown because the size of the data point exceeds the error bar."*

**m) Fig. 9:** How were the errors calculated? What do the small circles represent?

**Answer:** The boxplots in Fig. 9 represent the variability of wind speed and direction at each level during the hovering time of 60 s. The caption of Fig. 9 was adapted as follows:

*"Figure 9: Variability of wind direction (left) and speed (right) during 60 s hovering at 10, 25 and 50 m a.g.l. for flights between 19:32 UTC and 00:32 UTC in the night 21 to 22 July 2015. The blue box contains 50 % of the data and represents the interquartile range with the median as a black line. The dashed lines show maximum and minimum values in case those values are within the 1.5 interquartile range. Values outside this range (outliers) are represented with circles."*

**Reviewer #3**

First, based on the placement of the temperature and humidity sensors (on the arm of the multicoptor below one rotor; Fig. 1), it is very likely that these meteorological measurements were negatively impacted by rotor-wash. Indeed, the discontinuities in the potential temperature profiles at UAV sampling locations (Fig. 8) support this idea. I suggest placing the meteorological sensors closer to the methane inlet and away from the rotors. At the very least, this flaw in the method should be openly discussed and addressed in future studies. The authors should also comment on why the humidity data obtained from this sensor was not presented. Ideally, the authors could demonstrate using laboratory tests that the current flying geometry and sampling strategies do not adversely affect the either the methane measurements or the wind estimates.

**Answer:** Yes, the rotor downwash has an impact on the temperature and humidity sensors, but this position was chosen to ensure a continuous flow to make the sensors faster. To address this, we included a sentence in the discussion (p. 9, l. 27):

*"Since the thermocouple was placed below a rotor, discontinuities were found while hovering; the temperature measurement is rather representative for the volume around the multicopter than for a point. But this ensured a continuous flow around the sensor, which increased its*

*response time. For analysis, temperature was averaged for hovering at each level during the methane measurements."*

The humidity data were not shown in this study because the used SHT75 sensor was not fast enough for vertical profiles and always showed a hysteresis. Although the hovering time was long enough for the sensor to adapt to surrounding conditions during the methane measurements, humidity data were not shown. This was because this sensor is not appropriate for this kind of UAV-based measurements.

Concerning the suggested laboratory tests with regard to effects of the flying geometry and sampling strategy on methane measurements and wind estimates, this was already shown in other studies and addressed in this revision.

For methane measurements, we would like to refer to the comments o) and w) of Reviewer #1 and for wind estimation to comment c) of Reviewer #2.

The interpretation of the methane concentration gradients rely heavily on the interpretation of the meteorological conditions and changes in the surface layer with time. As a result, it would be helpful for figures 7 and 8 to show the local time, as well as UTC time.

**Answer:** Yes, we agree with this. In the manuscript, we included the sentence (p. 6, l. 19):

*"Time is given in UTC which corresponds to CEST-2."*

In addition, we stated local time in the caption of Fig. 7 and 8 explicitly:

Fig. 7: *"Local time (CEST) is UTC+2."*

Fig. 8: *"This corresponds to: 21:32 CEST to 02:32 CEST (UTC+2)."*

The discussion on L25-30 is difficult to follow and should be re-written.

**Answer:** Since the page number for this comment was not given, it was not clear to us which discussion was meant by the Reviewer. Therefore, unfortunately, we could not address it.

Finally, as noted in the manuscript, a more powerful UAV with a larger payload would enable longer profiles by ground-based gas spectrometer. Given a UAV with a larger payload, could the authors comment on what is the maximum altitude that could be reasonably sampled using this method, either due to prolonged residence time in the tubing, flow restrictions or other logistical concerns?

**Answer:** From our point of view, it is difficult to give a maximum altitude, which is possible with this method. We would rather suggest the other way round. (1) What is the altitude you want to reach, (2) how long you want to sample at each height and (3) at how many heights?

From (1) you know the length of your tube. The residence time in the tube is dependent on the pump in the gas analyzer and the diameter of the tube. This defines the possible flow rate and the sampling time at each height. Is (1) not reachable, a higher payload is needed or an adaptation of (2). Depending on the payload the possible flight time defines (3). In addition, it has to be highlighted that the material of the tube should not react with the gas of interest. Therefore, inert gases as methane are preferable to measure with this method.

From other methane measurements during the campaign we know that it is possible to use a 1/4 inch Teflon tube with a length of 140 m. But the weight of this tube is about 2 kg, which would have been too heavy for our hexacopter.

So, the existing parameters and limitations affect the possible reachable altitude for each individual application. Therefore, no maximum altitude was stated in the manuscript.

**Further technical corrections by authors**

a) "LiDAR" was replaced by "LIDAR" throughout the text

b) p. 2, l. 19: "size distribution" instead of "size distributions"

c) p. 3, l. 7: "TERrestrial" instead of "TERrestrail"

d) p. 3, l. 20: "Mauder et al., 2013" instead of "Mauder et al., 2014"

e) p. 4, l. 17: Comma was inserted

**Additional literature included in the references**

Alvarado, M., Gonzales, F., Erskine, P., Cliff, D. and Heuff, D.: A Methodology to Monitor Airborne $PM_{10}$ Dust Particles Using a Small Unmanned Aerial Vehicle, Sensors, 17, 343, doi:10.3390/s17020343, 2017.

Banta, R. M., Pichugina, Y. L., Kelley, N. D., Hardesty, R. M., and Brewer, W. A.: Wind energy meteorology: Insight into wind properties in the turbine-rotor layer of the atmosphere from high-resolution Doppler lidar, Bull. Amer. Meteorol. Soc., 94, 883-902, doi:http://dx.doi.org/10.1175/BAMS-D-11-00057.1, 2013.

Egger, J., Bajrachaya, S., Heinrich, R., Kolb, P., Lämmlein, S., Mech, M., Reuder, J., Schäper, W., Shakya, P., Schween, J., and Wendt, H.: Diurnal winds in the Himalayan Kali Gandaki valley. Part III: Remotely piloted aircraft soundings, Mon. Weather Rev, 130(8), 2042-2058, 2002.

Hammann, E., Behrendt, A., Le Mounier, F., and Wulfmeyer, V.: Temperature profiling of the atmospheric boundary layer with rotational Raman lidar during the HD(CP)2 Observational Prototype Experiment, Atmos. Chem. Phys., 15, 2867-2881, doi:10.5194/acp-15-2867-2015, 2015.

Holland G. J., McGeer T., and Youngren H.: Autonomous aerosondes for economical atmospheric soundings anywhere on the globe, Bull. Amer. Meteorol. Soc., 73(12), 1987-1998, 1992.

McGeer T., and Holland G.: Small autonomous aircraft for economical oceanographic observations on a wide scale, Oceanography, 6(3), 129-135, 1993.

Palomaki, R. T., Rose, N. T., van den Bossche, M., Sherman, T., J., and De Wekker, S. F. J.: Wind Estimation in the Lower Atmosphere Using Multirotor Aircraft, Atmos. Oceanic Technol., doi:10.1175/JTECH-D-16-0177.1, in press, 2017.

Rennó N. O., and Williams E. R.: Quasi-lagrangian measurements in convective boundary layer plumes and their implications for the calculation of cape. Mon. Weather Rev, 123(9): 2733-2742, 1995.

Soddell, J. R., McGuffie, K., and Holland, G. J.: Intercomparison of atmospheric soundings from the aerosonde and radiosondes, J. Appl. Meteor. Climatol., 43(9), 1260-1269, 2004.

Spiess T., Bange J., Buschmann M., and Vörsmann P.: First application of the meteorological Mini-UAV "M2AV", Meteorol. Z., 16(2), 159-169, doi:10.1127/0941-2948/2007/0195, 2007.

**Updated Figures and captions**

[Figure]

**Figure 1: Measurement site DE-Fen, Germany, with land use and ground-based instrumentation important for this study during the ScaleX campaign 2015. Contour lines stand for altitude (m) above sea level (QGIS, OpenStreetMap).**

[Figure]

**Figure 5: Wind direction (WD) and speed (WS) comparison between tower (grey) and multicopter (blue) at 9 m a.g.l. over 5 min. The colored bands around the lines represent the standard deviation of each time series.**

[Figure]

**Figure 6: Wind direction and speed profiles during two different flights: 09:01 UTC (left panels) and 09:31 UTC (right panels) on 15 July 2015. The blue profiles show multicopter data, dark grey circles represent EC station data, light grey squares LIDAR data and orange squares SODAR data. LIDAR and EC station data were averaged over the time the multicopter needed for the profile. Error bars show their standard deviation.**

[Figure]

**Figure 8: Vertical potential temperature ($T_{pot}$) profiles in blue and methane concentrations in green over six hours from 19:32 UTC (left) to 00:32 UTC (right) in the night 21 to 22 July 2015. This corresponds to: 21:32 CEST to 02:32 CEST (UTC+2). Air temperature was measured with the thermocouple (ascent data only) and $T_{pot}$ was calculated with the onboard pressure data from the autopilot. $T_{pot}$ was averaged at hovering levels and smoothed with a moving average (3 s). Error bars of methane concentration show the standard deviation for each measurement averaged over 60 s. A standard deviation of 0.01 ppm or less cannot be shown because the size of the data point exceeds the error bar.**